# Adaptive Exponential Decay Rates for Adam

## Abstract

Adam and its variants, including AdaBound, AdamW, and AdaBelief, have gained widespread popularity for enhancing the learning speed and generalization performance of deep neural networks. This optimization technique adjusts weight vectors by utilizing predetermined exponential decay rates (i.e.,$\beta_1 = 0.9$, $\beta_2 = 0.999$) based on the first moment estimate and the second raw moment estimate of the gradient. However, the default exponential decay rates might not be optimal, and the process of tuning them through trial and error with experience proves to be time-consuming. In this paper, we introduce AdamE, a novel variant of Adam designed to automatically leverage dynamic exponential decay rates on the first moment estimate and the second raw moment estimate of the gradient. Additionally, we provide theoretical proof of the convergence of AdamE in both convex and non-convex cases. To validate our claims, we perform experiments across various neural network architectures and tasks. Comparative analyses with adaptive methods utilizing default exponential decay rates reveal that AdamE consistently achieves rapid convergence and high accuracy in language modeling, node classification, and graph clustering tasks.

## 1 Introduction

Adam Kingma & Ba (2014), a widely adopted stochastic optimization method, has been applied across various domains in recent years, including object detection Kim et al. (2022), natural language processing Gu et al. (2022), and node classification Meng et al. (2021). Despite its widespread use, Adam's performance in training deep neural networks (DNNs) can be sensitive to improper learning rates, whether too large or too small. To address this issue, several improved variants have been proposed, such as AdaBound Luo et al. (2019), RAdam Liu et al. (2019), Padam Chen et al. (2018), AdamW Loshchilov & Hutter (2017), and AdaBelief Zhuang et al. (2020).

Building on the principles of adaptive optimization, we introduce a novel technique, AdamE, which incorporates adaptive exponential decay rates. This approach dynamically adjusts the decay rates for both the first and second moments of the gradient. The primary contributions of our study are as follows:

- **Introduction of AdamE**: We propose a new optimization algorithm, AdamE, designed for DNNs. Compared to the standard Adam optimizer and other mainstream methods, AdamE exhibits superior adaptability to current gradient values, resulting in faster convergence and improved model stability.

- **Theoretical analysis of AdamE**: We provide a detailed theoretical analysis of AdamE's convergence properties in both convex and non-convex stochastic optimization settings, demonstrating its enhanced convergence behavior.

- **Experimental validation**: The effectiveness of AdamE is validated through experiments on established datasets, including WikiText-2, BBBP, Cora, Citeseer, and DBLP. Extensive experimental results show that AdamE achieves state-of-the-art performance.

## 2 Motivation

Adam and its variants share a key feature: they rely on the first and second raw moment estimates of the gradient to optimize weight vectors, using default exponential decay rates (e.g., $\beta_1 = 0.9$,

$\beta_2 = 0.999$). However, there is limited research on how varying these decay rate settings affects Adam's convergence, generalization, and stability. In this study, we explore the effects of different combinations of exponential decay rates ($\beta_1 \in \{0.5, 0.7, 0.9\}$ and $\beta_2 \in \{0.9, 0.95, 0.999\}$) on Adam's performance in terms of these three aspects. The experimental findings are presented below:

A commonly used quadratic function is employed to assess the impact of different exponential decay rate settings on the convergence and stability of Adam. The function is defined as $f(x) = (x-1)^2 + 2$, with a search domain of $-5 \le x \le 7$. The global minimum of the quadratic function is at $\tilde{x} = 1$, $f(\tilde{x}) = 2$. The learning rate and number of iterations for Adam with different ($\beta_1, \beta_2$) are set to 0.8 and 200 respectively.

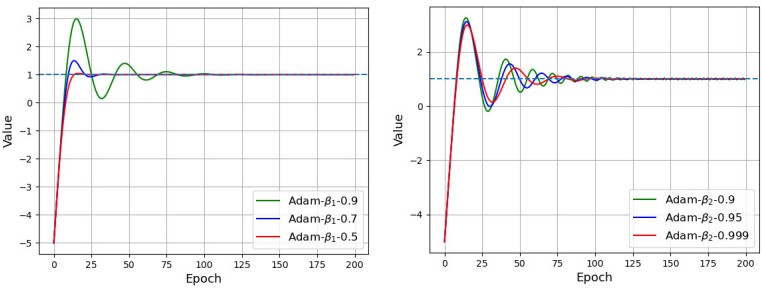

(a) Adam with different $\beta_1$      (b) Adam with different $\beta_2$

Figure 1: Results of Adam with different $\beta_1$ and $\beta_2$ for Quadratic function.

As depicted in Fig. 1(a), as the $\beta_1$ of Adam increasing (with $\beta_2$ fixed at 0.999), the overshoot in function optimization for Adam gradually increase, and the convergence speed gradually decreases. The fluctuating range of the function optimization curve is higher when $\beta_1$ =0.9. In comparison with Fig. 1(a), the result of Fig. 1(b) is different, the fluctuating range of function optimization curve gradually decreases with an increase in $\beta_2$ of Adam (with $\beta_1$ fixed at 0.9). The optimal function optimization results for Adam can be observed when $\beta_1$=0.5 and $\beta_2$=0.999.

The experimental outcomes for Adam, considering different ($\beta_1$, $\beta_2$) combinations, across tasks such as quadratic function, emphasize the critical role of appropriately setting $\beta_1$ and $\beta_2$ for Adam based on specific tasks in training DNNs. In this paper, we introduce a novel variant of Adam, termed Adam with dynamic exponential decay rate (AdamE), which straightforwardly computes $\beta_1$ and $\beta_2$ of Adam based on iterations.

## 3 ALGORITHM

### 3.1 DETAILS OF ADAME

The pseudo-code for AdamE is presented in Algorithm 1. In comparison to Adam, AdamE exhibits the capability to dynamically adjust the exponential decay rates of the first moment estimate ($\alpha$) and the second raw moment estimate ($\beta$) based on the timestep. Essentially, AdamE flexibly adapts its step size in accordance with the exponential decay rates of the first moment estimate and the second raw moment estimate, all without the need for bias correction.

---

**Algorithm 1** AdamE

---

**Input:** $\theta_0$ is initial parameter vector, the good default settings are $\lambda = 0.01$, $\varepsilon = 10^{-8}$, $d_0 = 0$, $s_0 = 0$ and $q = 0$.
**Output:** The parameters $\theta_T$ of the model.
**for** $q = 1$; $q \le T$ **do**
    $g_q = \nabla_\theta f_q(\theta_{q-1})$; (Get gradients of stochastic objective function at $q$th epoch)
    $\alpha_q = \frac{q}{1+q^2}$; (Calculate exponential decay rate of first moment estimate)
    $\beta_q = 1 - \alpha_q$; (Calculate exponential decay rate of second raw moment estimate)
    $d_q = \alpha_q * d_{q-1} + (1 - \alpha_q) * g_q$; (Calculate first moment estimate)
    $s_q = \beta_q * s_{q-1} + (1 - \beta_q) * g_q^2$; (Calculate second raw moment estimate)
    $\theta_q = \theta_{q-1} - \lambda * \frac{d_q}{\sqrt{s_q} + \varepsilon}$ (Update parameters of model)
**end for**

---

### 3.2 NUMERICAL EXPERIMENTS

To assess the efficacy of AdamE, depicted in Fig. 1(a) and Fig. 1(b), an experiment involving quadratic function fitting was conducted using AdamE. The outcomes, illustrated in Fig. 2(a) and

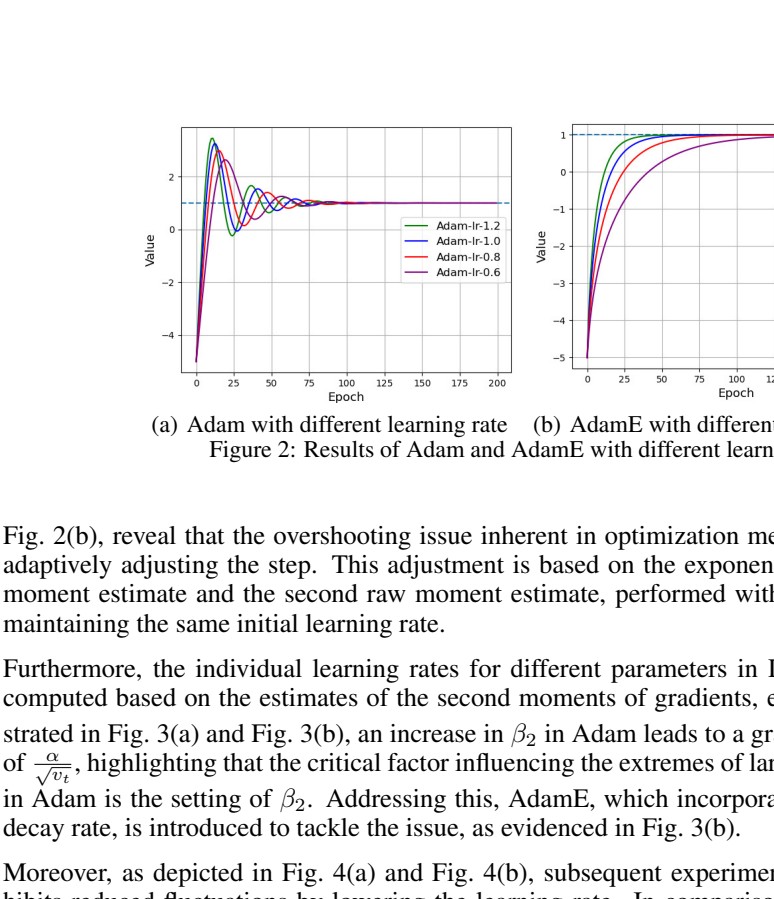

(a) Adam with different learning rate  (b) AdamE with different learning rate

Figure 2: Results of Adam and AdamE with different learning rate.

Fig. 2(b), reveal that the overshooting issue inherent in optimization methods can be mitigated by adaptively adjusting the step. This adjustment is based on the exponential decay rates of the first moment estimate and the second raw moment estimate, performed without bias correction, while maintaining the same initial learning rate.

Furthermore, the individual learning rates for different parameters in DNNs can be dynamically computed based on the estimates of the second moments of gradients, expressed as $\frac{\alpha}{\sqrt{v_t}}$. Demonstrated in Fig. 3(a) and Fig. 3(b), an increase in $\beta_2$ in Adam leads to a gradual decrease in the value of $\frac{\alpha}{\sqrt{v_t}}$, highlighting that the critical factor influencing the extremes of large and small learning rates in Adam is the setting of $\beta_2$. Addressing this, AdamE, which incorporates a dynamic exponential decay rate, is introduced to tackle the issue, as evidenced in Fig. 3(b).

Moreover, as depicted in Fig. 4(a) and Fig. 4(b), subsequent experiments indicate that Adam exhibits reduced fluctuations by lowering the learning rate. In comparison, AdamE proves adept at overcoming the overshooting problem; however, its convergence speed gradually diminishes with decreasing learning rates. Consequently, future investigations are directed towards enhancing the convergence s

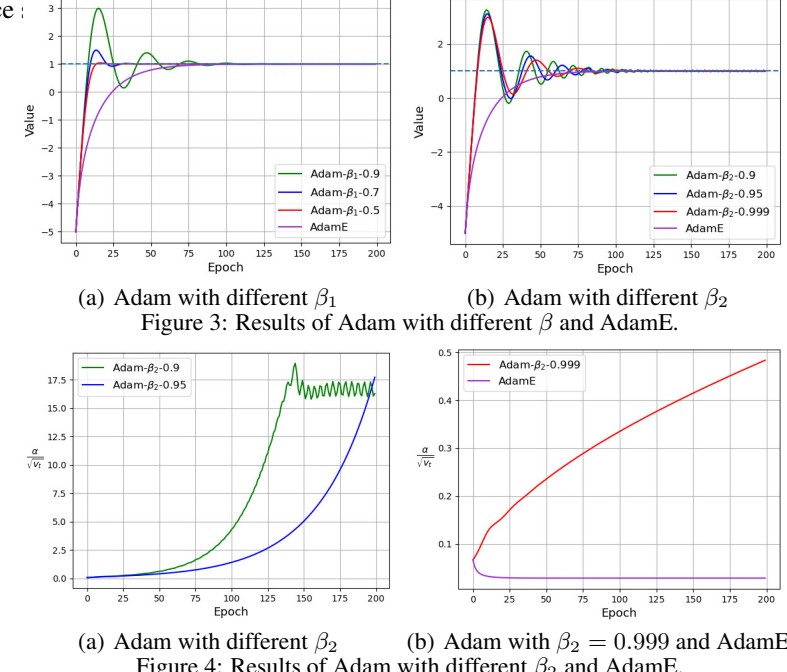

(a) Adam with different $\beta_1$  (b) Adam with different $\beta_2$

Figure 3: Results of Adam with different $\beta$ and AdamE.

(a) Adam with different $\beta_2$  (b) Adam with $\beta_2 = 0.999$ and AdamE

Figure 4: Results of Adam with different $\beta_2$ and AdamE.

### 3.3 ANALYSIS OF THE REASONS WHY ADAME CAN IMPROVE THE CONVERGENCE OF ADAM IN THE CONVEX CASE

To demonstrate the improvements AdamE offers over Adam in terms of convergence, we analyze the convergence behavior of AdamE in the convex setting, as detailed below.

**Theorem 1.1** Kingma & Ba (2014): Assuming that the function $f_t$ has bounded gradients, $\|\nabla f_t(\theta_t)\|_2 \leq G$, $\|\nabla f_t(\theta_t)\|_\infty \leq G_\infty$ for all $\theta \in R^d$ and distance between any $\theta_t$ generated by Adam is bounded as $\|\theta_n - \theta_m\|_2 \leq D$, $\|\theta_n - \theta_m\|_\infty \leq D_\infty$ for any $m, n \in 1, ..., T$, and $\beta_1$, $\beta_2 \in [0, 1)$ satisfy $\frac{\beta_1^2}{\sqrt{\beta_2}} < 1$. Let $\alpha_t = \frac{\alpha}{\sqrt{t}}$ and $\beta_{1,t} = \beta_1 \lambda^{t-1}$, $\lambda \in (0, 1)$. Adam achieves the following guarantee, for all $T \geq 1$.

$$R(T) = \frac{D^2}{2\alpha(1-\beta_1)} \sum_{i=1}^{d} \sqrt{T\hat{v}_{T,i}} + \frac{\alpha(\beta_1+1)G_\infty \sum_{i=1}^{d} \|g_{1:T,i}\|_2}{(1-\beta_1)\sqrt{1-\beta_2}(1-\gamma)^2} + \sum_{i=1}^{d} \frac{D_\infty^2 G_\infty \sqrt{1-\beta_2}}{2\alpha(1-\beta_1)(1-\lambda)^2} \quad (1)$$

According to literature Kingma & Ba (2014), we have

$$F_{t+1} = \frac{\sqrt{\hat{v}_{t+1}}}{\tau_{t+1}} - \frac{\sqrt{\hat{v}_t}}{\tau_t} > \frac{\sqrt{\beta_2 \hat{v}_t}}{\tau_{t+1}} - \frac{\sqrt{\hat{v}_t}}{\tau_t} > \frac{\sqrt{\hat{v}_t}}{\tau}(\sqrt{\beta_2(t+1)} - \sqrt{t}) \quad (2)$$

The core of convergence of Adam is $F_{t+1} = \frac{\sqrt{v_{t+1}}}{\tau_{t+1}} - \frac{\sqrt{v_t}}{\tau_t} > 0$, which $v_t = \beta_2 v_{t-1} + (1-\beta_2)g_t^2 = (1-\beta_2)\sum_{i=1}^{t} \beta_2^{t-i} g_i^2$ is the second moment estimate of Adam, and $\tau_t = \frac{\tau}{\sqrt{t}}$ ($\tau$ is the initial learning rate of Adam). However, according to Theorem 1.1, if $0 < \beta_2 < \frac{t}{t+1} < 1$, and $\beta_2(t+1) < t$, so it is hard to illustrate $F_{t+1} = \frac{\sqrt{v_{t+1}}}{\tau_{t+1}} - \frac{\sqrt{v_t}}{\tau_t} > 0$. However, when $\frac{t}{t+1} < \beta_2 < 1$ and $\beta_2(t+1) > t$, it ensures that $F_{t+1} = \frac{\sqrt{v_{t+1}}}{\tau_{t+1}} - \frac{\sqrt{v_t}}{\tau_t} > 0$, so the value of $\beta_2$ should be as close as possible to 1, such as $\beta_2 = 0.999$.

According to Algorithm 1, we have

$$H_{t+1} = \frac{\sqrt{s_{t+1}}}{\tau_{t+1}} - \frac{\sqrt{s_t}}{\tau_t} > \frac{\sqrt{\beta_{t+1} s_t}}{\tau_{t+1}} - \frac{\sqrt{s_t}}{\tau_t} > \frac{\sqrt{s_t}}{\tau}(\sqrt{\beta_{t+1}(t+1)} - \sqrt{t}) \quad (3)$$

Owing to

$$\sqrt{\beta_{t+1}(t+1)} - \sqrt{t} = \sqrt{(1-\alpha_{t+1})(t+1)} - \sqrt{t} = \sqrt{(t+1) - \frac{(t+1)^2}{(t+1)^2 + 1}} - \sqrt{t} > 0 \quad (4)$$

Since $H_{t+1} > 0$, AdamE exhibits the capability to improve the convergence of Adam for convex scenarios.

# 4 CONVERGENCE ANALYSIS FOR CONVEX CASE

The regret is defined as

$$R(T) = \sum_{t=1}^{T} [f_t(\theta_t) - f_t(\theta^*)] \quad (5)$$

where $\theta^* = \arg\min_{\theta \in \chi} \sum_{t=1}^{T} f_t(\theta_t)$. Some definitions of mathematical symbols are represented as $g_t = \nabla f_t(\theta_t)$, $g_{t,i}$ is the $i^{th}$ element, $g_{1:t,i} \in \mathbb{R}^t$ is a vector which includes the $i^{th}$ dimension of gradients $g_t$ over all iterations till $t$ as $g_{1:t,i} = [g_{1,i}, g_{2,i}, ..., g_{t,i}]$.

Online learning framework Kingma & Ba (2014) is applied in convergence analysis of AdamE for convex case, and we prove that AdamE has a regret bound using the following theorems.

**Lemma 2.1** : If a function $f : R^d \to R$ is convex, then for all $x, y \in R^d$,

$$f(y) \geq f(x) + \nabla f(x)^T(y - x) \quad (6)$$

**Theorem 2.1**: Let $g_t = \nabla f_t(\theta_t)$ and $g_{1:t}$ are defined as bounded and above, $\|g_t\|_2 \leq G_2$. Then,

$$\sum_{t=1}^{T} \frac{m_{t,i}^2}{\sqrt{t v_{t,i}}} < \frac{16\sqrt{T}}{(\sqrt{8}-1)^2} \|g_{1:T,i}\|_2 \quad (7)$$

*Proof of Theorem 2.1.* According to Algorithm 1, we have

$$m_{t,i} = \alpha_t m_{t-1,i} + (1 - \alpha_t)g_{t,i} = \sum_{i=1}^{t}(1 - \alpha_i)g_{t,i}\prod_{k=i+1}^{t}\alpha_k \tag{8}$$

$$v_{t,i} = \beta_t v_{t-1,i} + (1 - \beta_t)g_{t,i}^2 = \sum_{i=1}^{t}(1 - \beta_i)g_{t,i}^2\prod_{k=i+1}^{t}\beta_k \tag{9}$$

Owing to $\alpha_t = \frac{t}{1+t^2}$ and $\beta_t = 1 - \alpha_t$, there are $\prod_{k=i+1}^{t}\alpha_k = \prod_{k=i+1}^{t}\frac{k}{1+k^2} \le 2^{i-t}$ and $\sqrt{\prod_{k=i+1}^{t}\beta_k} = \sqrt{\prod_{k=i+1}^{t}(1 - \alpha_k)} \ge \sqrt{(\frac{1}{2})^{t-i}} = 2^{\frac{i-t}{2}}$.

We expand the last term in the summation of Algorithm 1,

$$\sum_{t=1}^{T}\frac{m_{t,i}^2}{\sqrt{tv_{t,i}}} = \sum_{t=1}^{T-1}\frac{m_{t,i}^2}{\sqrt{tv_{t,i}}} + \frac{(\sum_{j=1}^{T}(1-\alpha_j)g_{j,i}\prod_{k=j+1}^{T}\alpha_k)^2}{\sqrt{T\sum_{j=1}^{T}(1-\beta_j)g_{j,i}^2\prod_{k=j+1}^{T}\beta_k}} < \sum_{t=1}^{T-1}\frac{m_{t,i}^2}{\sqrt{tv_{t,i}}} + \frac{(\sum_{j=1}^{T}(1-\alpha_j)g_{j,i}2^{j-T})^2}{\sqrt{T\sum_{j=1}^{T}(1-\beta_j)g_{j,i}^2 2^{j-T}}}$$

$$< \sum_{t=1}^{T-1}\frac{m_{t,i}^2}{\sqrt{tv_{t,i}}} + \sum_{j=1}^{T}\frac{T((1-\alpha_j)g_{j,i}2^{j-T})^2}{\sqrt{T\sum_{j=1}^{T}(1-\beta_j)g_{j,i}^2 2^{j-T}}} < \sum_{t=1}^{T-1}\frac{m_{t,i}^2}{\sqrt{tv_{t,i}}} + \sum_{j=1}^{T}\frac{T((1-\alpha_j)g_{j,i}2^{j-T})^2}{\sqrt{T(1-\beta_j)g_{j,i}^2 2^{j-T}}}$$

$$< \sum_{t=1}^{T-1}\frac{m_{t,i}^2}{\sqrt{tv_{t,i}}} + \sum_{j=1}^{T}\sqrt{T}\sqrt{\frac{(1-\alpha_j)^2}{\alpha_j}}\frac{2^{2j-2T}}{\sqrt{2^{j-T}}}\frac{g_{j,i}^2}{\sqrt{g_{j,i}^2}} < \sum_{t=1}^{T-1}\frac{m_{t,i}^2}{\sqrt{tv_{t,i}}} + \sum_{j=1}^{T}\sqrt{T}(\frac{1}{\sqrt{\alpha_j}} - \sqrt{\alpha_j})(2\sqrt{2})^{j-T}\|g_{j,i}\|_2$$

$$< \sum_{t=1}^{T-1}\frac{m_{t,i}^2}{\sqrt{tv_{t,i}}} + \sqrt{T}\sum_{j=1}^{T}\frac{(2\sqrt{2})^{j-T}}{\sqrt{\alpha_j}}\|g_{j,i}\|_2. \tag{10}$$

So we get that

$$\sum_{t=1}^{T}\frac{m_{t,i}^2}{\sqrt{tv_{t,i}}} < \sum_{t=1}^{T}\frac{\|g_{t,i}\|_2}{\sqrt{\alpha_t}}\sum_{k=0}^{T-t}\sqrt{t}(\frac{1}{2\sqrt{2}})^k. \tag{11}$$

For $\frac{1}{2\sqrt{2}} < 1$, using the upper bound on the arithmetic-geometric series,

$$\sum_{t}\sqrt{t}(\frac{1}{2\sqrt{2}})^t < \frac{1}{(1-\frac{1}{2\sqrt{2}})^2} = \frac{8}{(\sqrt{8}-1)^2}, \tag{12}$$

then, we have

$$\sum_{t=1}^{T}\frac{m_{t,i}^2}{\sqrt{tv_{t,i}}} < \sum_{t=1}^{T}\frac{\|g_{t,i}\|_2}{\sqrt{\alpha_t}}\sum_{k=0}^{T-t}\sqrt{t}(\frac{1}{2\sqrt{2}})^k < \frac{8}{(\sqrt{8}-1)^2}\sum_{t=1}^{T}\frac{\|g_{t,i}\|_2}{\sqrt{\alpha_t}}$$

$$< \frac{16}{(\sqrt{8}-1)^2}\sum_{t=1}^{T}\sqrt{t}\|g_{t,i}\|_2 < \frac{16\sqrt{T}}{(\sqrt{8}-1)^2}\|g_{1:T,i}\|_2. \tag{13}$$

**Theorem 2.2**: Given a cost function $f_t$ with bounded gradients, which is $\|g_t\|_2 = \|\nabla f_t(\theta_t)\|_2 \le G_2$. For all $\theta \in R^d$ and distance between any $\theta_t$ generated by AdamE is bounded as $\|\theta_n - \theta_m\|_2 \le D_2$ for any $m, n \in 1, ..., T$. Let $\lambda_t = \frac{\lambda}{t}$, AdamE achieves the following guarantee, for all $T \ge 1$.

$$R(T) < \frac{D_2^2}{\lambda}\sum_{i=1}^{d}\sqrt{Tv_{T,i}} + \lambda\sum_{i=1}^{d}\frac{16\sqrt{T}}{(\sqrt{8}-1)^2}\|g_{1:T,i}\|_2 + D_2\sum_{i=1}^{d}m_{T-1,i} \tag{14}$$

It is shown that the regret of AdamE is upper bounded by $O(\sqrt{T})$, which is similar to Adam and its variants.

*Proof of Theorem 2.2.* According to Lemma 2.1, we have

$$f_t(\theta_t) - f_t(\theta^*) \le g_t^T(\theta_t - \theta^*). \tag{15}$$

According to Algorithm 1:

$$\theta_{t+1} = \theta_t - \lambda_t \frac{m_t}{\sqrt{v_t}} = \theta_t - \lambda_t (\frac{\alpha_t m_{t-1}}{\sqrt{v_t}} + \frac{(1-\alpha_t)g_t}{\sqrt{v_t}}). \tag{16}$$

The $i^{th}$ dimension of parameter vector $\theta_t \in R^d$ is a particular focus of attention. Subtract $\theta^*$ and square both side of above formula, we get

$$
\begin{aligned}
(\theta_{t+1,i} - \theta_i^*)^2 &= (\theta_{t,i} - \theta_i^* - \lambda_t \frac{m_{t,i}}{\sqrt{v_{t,i}}})^2 \\
&= (\theta_{t,i} - \theta_i^*)^2 + \lambda_t^2 (\frac{m_{t,i}}{\sqrt{v_{t,i}}})^2 - 2\lambda_t (\frac{\alpha_t m_{t-1,i}}{\sqrt{v_{t,i}}} + \frac{(1-\alpha_t)g_{t,i}}{\sqrt{v_{t,i}}})(\theta_{t,i} - \theta_i^*), \quad (17)
\end{aligned}
$$

so we have $\sqrt{v_{t,i}} = \sqrt{\sum_{i=1}^t (1-\beta_i)g_{t,i}^2 \prod_{k=i+1}^t \beta_k} < \|g_{1:t,i}\|_2$.

$$
\begin{aligned}
g_{t,i}(\theta_{t,i} - \theta_i^*) &= \frac{\sqrt{v_{t,i}}}{2\lambda_t(1-\alpha_t)}((\theta_{t,i} - \theta_i^*)^2 - (\theta_{t+1,i} - \theta_i^*)^2) + \frac{\lambda_t}{2(1-\alpha_t)}\frac{m_{t,i}^2}{\sqrt{v_{t,i}}} + \frac{\alpha_t}{1-\alpha_t}m_{t-1,i}(\theta_{t,i} - \theta_i^*) \\
&< \frac{\sqrt{v_{t,i}}}{2\lambda_t(1-\alpha_t)}(\theta_{t,i} - \theta_i^*)^2 + \frac{\lambda_t}{2(1-\alpha_t)}\frac{m_{t,i}^2}{\sqrt{v_{t,i}}} + \frac{\alpha_t}{1-\alpha_t}m_{t-1,i}(\theta_{t,i} - \theta_i^*) \\
&< \frac{\sqrt{v_{t,i}}}{\lambda_t}(\theta_{t,i} - \theta_i^*)^2 + \lambda_t \frac{m_{t,i}^2}{\sqrt{v_{t,i}}} + m_{t-1,i}(\theta_{t,i} - \theta_i^*). \tag{18}
\end{aligned}
$$

For the sequence of convex functions $f_t(\theta_t)$ ($t \in 1, ..., T$), the upper regret bound of $f_t(\theta_t) - f_t(\theta^*)$ by summing across all dimension for $i \in 1, ..., d$ can be obtained by using Lemma 2.1 to above inequality.

$$R(T) < \sum_{t=1}^T \sum_{i=1}^d (\frac{\sqrt{tv_{t,i}}}{\lambda}(\theta_{t,i} - \theta_i^*)^2 + \lambda \frac{m_{t,i}^2}{\sqrt{tv_{t,i}}} + m_{t-1,i}(\theta_{t,i} - \theta_i^*)). \tag{19}$$

According to Theorem 2.1, $\sum_{t=1}^T \frac{m_{t,i}^2}{\sqrt{tv_{t,i}}} < \frac{16\sqrt{T}}{(\sqrt{8}-1)^2}\|g_{1:T,i}\|_2$, and $\|\theta_n - \theta_m\|_2 \le D_2$, we have:

$$R(T) < \frac{D_2^2}{\lambda}\sum_{i=1}^d \sqrt{Tv_{T,i}} + \lambda \sum_{i=1}^d \frac{16\sqrt{T}}{(\sqrt{8}-1)^2}\|g_{1:T,i}\|_2 + D_2 \sum_{i=1}^d m_{T-1,i}. \tag{20}$$

## 5 CONVERGENCE ANALYSIS FOR NONCONVEX CASE

According to Zhou et al. (2018), we present theoretical results on the convergence of AdamE. The following stochastic nonconvex optimization problem will be further studied

$$\min_{x\in\mathbb{R}^d} f(x) := \mathbb{E}_\xi[f(x;\xi)], \tag{21}$$

where $\xi$ is a random variable satisfying certain distribution, and $f(x;\xi) : \mathbb{R}^d \longrightarrow R$ is a L-smooth nonconvex function. $x_* \in \arg\min_{x\in\mathbb{R}^d} f(x) := \mathbb{E}_\xi[f(x;\xi)]$ exits.

**Lemma 3.1**: $f(x) = \mathbb{E}_\xi[f(x;\xi)]$ is L-smooth, which has a $G_2$-bounded stochastic gradient. That is, for any $\xi$, $x$ and $y$ ($x, y \in \mathbb{R}^d$), we have

$$\|\nabla f(x;\xi)\|_2 \le G_2, \tag{22}$$

$$|f(x) - f(y) - \langle \nabla f(y), x-y \rangle| \le \frac{L}{2}\|x-y\|_2^2. \tag{23}$$

$m_t$ and $v_t$ can be as defined in Algorithm 1. Then under Assumption 3.1, we have $\|v_t\|_2 \le G_2^2$ and $\|m_t\|_2 \le G_2$.

**Theorem 3.1**: According to Algorithm 1, let $\alpha_t$ and $\beta_t$ be the weight parameters such as $\alpha_t = \frac{t}{1+t^2}$, $\beta_t = 1 - \alpha_t$, $m_t = \alpha_t m_{t-1} + (1 - \alpha_t)g_t$ and $v_t = \beta_t v_{t-1} + (1 - \beta_t)g_t^2$, $\lambda_t, t = 1, ..., T$ is the step sizes. Suppose that $\lambda_t = \frac{\lambda}{\sqrt{t}}$, then under Assumption 3.1, we have the following results:

$$\frac{\sum_{t=1}^{T} \mathbb{E}[\|\nabla f(x_t;\xi)\|^2]}{T} < \frac{G_2}{\lambda\sqrt{T}}(\frac{G_2^2 d}{4L} + f(x_1;\xi) - f(x_*) + 2\lambda G_2^2 \|v_0^{-1/2}\|_2 + 4dL\lambda^2(1 + \log T)) \tag{24}$$

Theorem 3.1 implies the convergence rate of AdamE in the non-convex case is upper bounded by $O(\log \frac{T}{\sqrt{T}})$.

*Proof of Theorem 3.1.* Let $A_t = \lambda_t v_t^{-1/2}\nabla f(x_t;\xi)$ for $t \geq 1$ and $A_0 = A_1$. Consider the definition of $m_t$ and $A_t \in \mathbb{R}^d$, $\forall t = 1, ..., T$. Then it follows that

$$\langle A_t, g_t \rangle = \frac{1}{1-\alpha_t}(\langle A_t, m_t \rangle - \langle A_{t-1}, m_{t-1} \rangle) + \langle A_{t-1}, m_{t-1} \rangle + \frac{\alpha_t}{1-\alpha_t}\langle A_{t-1} - A_t, m_{t-1} \rangle \tag{25}$$

We use $A_t = \lambda_t v_t^{-1/2}\nabla f(x_t;\xi)$. By summing (25) over $t = 1, ..., T$ and using the initial condition $m_0 = 0$, we get

$$\sum_{t=1}^{T} \langle A_t, g_t \rangle = \frac{1}{1-\alpha_t}\langle A_T, m_T \rangle + \sum_{t=1}^{T-1} \langle A_t, m_t \rangle + \frac{\alpha_t}{1-\alpha_t}\sum_{t=1}^{T}\langle A_{t-1} - A_t, m_{t-1} \rangle$$
$$= \frac{\alpha_t}{1-\alpha_t}\langle A_T, m_T \rangle + \sum_{t=1}^{T}\langle A_t, m_t \rangle + \frac{\alpha_t}{1-\alpha_t}\sum_{t=1}^{T-1}\langle A_t - A_{t+1}, m_t \rangle \tag{26}$$

For(26), the goal is to derive bounds and calculate expectation to estimation $\mathbb{E}\left[\|\nabla f(x_t;\xi)\|^2\right]$. The intelligible idea is to calculate it by using $\mathbb{E}[\langle A_t, g_t \rangle] = \mathbb{E}\left[\left\langle \lambda_t v_t^{-1/2}\nabla f(x_t;\xi), g_t \right\rangle\right]$. So we need a more suitable random variable to take place $\langle A_t, g_t \rangle$ for taking conditional expectation $\mathbb{E}_t$.

Bound for $\langle A_t, g_t \rangle$

$$\langle A_t, g_t \rangle = \left\langle \lambda_t v_t^{-1/2}\nabla f(x_t;\xi), g_t \right\rangle$$
$$= \left\langle \lambda_{t-1} v_{t-1}^{-1/2}\nabla f(x_t;\xi), g_t \right\rangle - \left\langle \nabla f(x_t;\xi), \left(\lambda_{t-1} v_{t-1}^{-1/2} - \lambda_t v_t^{-1/2}\right) g_t \right\rangle. \tag{27}$$

We set that $\lambda_0 = \lambda = \lambda_1$ for simplifying derivations. Now for the last term in the right-hand side, by using Hölder's inequality, and $\lambda_{t-1} v_{t-1,i} \geq \lambda_t v_{t,i}$ (note that for $t = 1$, this is still true), we have

$$\left\langle \nabla f(x_t;\xi), \left(\lambda_{t-1} v_{t-1}^{-1/2} - \lambda_t v_t^{-1/2}\right) g_t \right\rangle < \|\nabla f(x_t;\xi)\|_2 \|\lambda_{t-1} v_{t-1}^{-1/2} - \lambda_t v_t^{-1/2}\|_2 \|g_t\|_2$$
$$< G_2^2 \left(\|\lambda_{t-1} v_{t-1}^{-1/2}\|_2 - \|\lambda_t v_t^{-1/2}\|_2\right), \tag{28}$$

Combining (27) and (28) yields

$$\langle A_t, g_t \rangle < \left\langle \lambda_{t-1} v_{t-1}^{-1/2}\nabla f(x_t;\xi), g_t \right\rangle - G_2^2 \left(\left\|\lambda_{t-1} v_{t-1}^{-1/2}\right\|_2 - \left\|\lambda_t v_t^{-1/2}\right\|_2\right) \tag{29}$$

Rearrange (29), we have

$$\left\langle \lambda_{t-1} v_{t-1}^{-1/2}\nabla f(x_t;\xi), g_t \right\rangle < \langle A_t, g_t \rangle + G_2^2 \left(\left\|\lambda_{t-1} v_{t-1}^{-1/2}\right\|_2 - \left\|\lambda_t v_t^{-1/2}\right\|_2\right). \tag{30}$$

It is obvious that the term $\left\langle \lambda_{t-1} v_{t-1}^{-1/2}\nabla f(x_t;\xi), g_t \right\rangle$ is more suitable for calculating $\mathbb{E}_t$. The next step is to focus on the bound of $\sum_{t=1}^{T}\langle A_t, g_t \rangle$ and consider each term separately.

Bound for $\langle A_T, m_T \rangle$: By Young's inequality, $x_T - x_{T-1} = \lambda_T v_T^{-1/2} m_T$, and $\|\nabla f(x_T;\xi)\|_2 \leq G_2$,

$$\langle A_T, m_T \rangle = \left\langle \nabla f(x_T;\xi), \lambda_T v_T^{-1/2} m_T \right\rangle < L\left\|\lambda_T v_T^{-1/2} m_T\right\|_2^2 + \frac{1}{4L}\|\nabla f(x_T;\xi)\|_2^2 < L\|x_{T+1} - x_T\|_2^2 + \frac{G_2^2}{4L} \tag{31}$$

**Bound for $\langle A_t, m_t \rangle$:** By the update of $x_{t+1}$, we have

$$\langle A_t, m_t \rangle = \langle \lambda_t v_t^{-1/2} \nabla f(x_t; \xi), m_t \rangle = \langle \nabla f(x_t; \xi), \lambda_t v_t^{-1/2} m_t \rangle$$
$$= \langle \nabla f(x_t; \xi), x_t - x_{t+1} \rangle < f(x_t; \xi) - f(x_{t+1}; \xi) + \frac{L}{2} \|x_{t+1} - x_t\|_2^2. \tag{32}$$

**Bound for $\langle A_t - A_{t+1}, m_t \rangle$**

$$\langle A_t - A_{t+1}, m_t \rangle = \left\langle \lambda_t v_t^{-1/2} \nabla f(x_t; \xi) - \lambda_{t+1} v_{t+1}^{-1/2} \nabla f(x_{t+1}; \xi), m_t \right\rangle$$

$$= \left\langle \lambda_t v_t^{-1/2} \nabla f(x_{t+1}; \xi) - \lambda_{t+1} v_{t+1}^{-1/2} \nabla f(x_{t+1}; \xi), m_t \right\rangle + \left\langle \lambda_t v_t^{-1/2} \nabla f(x_t; \xi) - \lambda_t v_t^{-1/2} \nabla f(x_{t+1}; \xi), m_t \right\rangle$$

$$= \left\langle \nabla f(x_{t+1}; \xi), \left( \lambda_t v_t^{-1/2} - \lambda_{t+1} v_{t+1}^{-1/2} \right) m_t \right\rangle + \left\langle \nabla f(x_t; \xi) - \nabla f(x_{t+1}; \xi), \lambda_t v_t^{-1/2} m_t \right\rangle \tag{33}$$

For the first term of (33), it uses similar derivation in (28), then we get

$$\left\langle \nabla f(x_{t+1}; \xi), \left( \lambda_t v_t^{-1/2} - \lambda_{t+1} v_{t+1}^{-1/2} \right) m_t \right\rangle < \|\nabla f(x_{t+1}; \xi)\|_2 \|\lambda_t v_t^{-1/2} - \lambda_{t+1} v_{t+1}^{-1/2}\|_2 \|m_t\|_2$$
$$< \|\nabla f(x_{t+1}; \xi)\|_2 \|\lambda_t v_t^{-1/2} - \lambda_{t+1} v_{t+1}^{-1/2}\|_2 \|g_t\|_2 < G_2^2 \left( \|\lambda_t v_t^{-1/2}\|_2 - \|\lambda_{t+1} v_{t+1}^{-1/2}\|_2 \right), \tag{34}$$

where $\|m_t\|_2 < \|g_t\|_2$ comes from $\|m_t\|_2 = \|\alpha_t m_{t-1} + (1 - \alpha_t) g_t\|_2 < \alpha_t \|m_{t-1}\|_2 + (1 - \alpha_t) \|g_t\|_2 < \alpha_t \|m_t\|_2 + (1 - \alpha_t) \|g_t\|_2$.

Because of $f(x; \xi)$ is L-smooth, then the update rule of $x_{t+1}$ as shown that

$$\left\langle \nabla f(x_t; \xi) - \nabla f(x_{t+1}; \xi), \lambda_t v_t^{-1/2} m_t \right\rangle < \|\nabla f(x_t; \xi) - \nabla f(x_{t+1}; \xi)\|_2 \|\lambda_t v_t^{-1/2} m_t\|_2$$
$$< L \|x_{t+1} - x_t\|_2 \|\lambda_t v_t^{-1/2} m_t\|_2 = L \|x_{t+1} - x_t\|_2^2. \tag{35}$$

Combining (34) and (35), we can derive

$$\langle A_t - A_{t+1}, m_t \rangle < G_2^2 \left( \left\| \lambda_t v_t^{-1/2} \right\|_2 - \left\| \lambda_{t+1} v_{t+1}^{-1/2} \right\|_2 \right) + \|x_{t+1} - x_t\|_2^2. \tag{36}$$

We then get that

$$\sum_{t=1}^T \langle A_t, g_t \rangle \leq \frac{G_2^2 d}{4L} + \left( f(x_1; \xi) - f(x_{T+1}; \xi) + \frac{L}{2} \sum_{t=1}^T \|x_{t+1} - x_t\|_2^2 \right) + G_2^2 \left\| \lambda_1 v_1^{-1/2} \right\|_2 + L \sum_{t=1}^T \|x_{t+1} - x_t\|_2^2$$

$$< \frac{G_2^2 d}{4L} + (f(x_1; \xi) - f(x_*)) + \lambda G_2^2 \left\| v_1^{-1/2} \right\|_2 + 2L \sum_{t=1}^T \|x_{t+1} - x_t\|_2^2 \tag{37}$$

where the last inequality follows from $f(x_{T+1}; \xi) \geq f(x_*), \alpha_1 = \alpha$.

Considering the last term of (37):

$$2L \sum_{t=1}^T \|x_{t+1} - x_t\|_2^2 = 2L \sum_{t=1}^T \left\| \lambda_t v_t^{-1/2} m_t \right\|_2^2 = 2L \sum_{t=1}^T \sum_{i=1}^d \lambda_t^2 \frac{m_{t,i}^2}{v_{t,i}}$$

$$= 2L \sum_{t=1}^T \sum_{i=1}^d \lambda_t^2 \frac{\left( \sum_{j=1}^t (1 - \alpha_j) g_{j,i} \prod_{k=j+1}^t \alpha_k \right)^2}{\sum_{j=1}^t (1 - \beta_j) g_{j,i}^2 \prod_{k=j+1}^t \beta_k} < 2L \sum_{t=1}^T \sum_{i=1}^d \lambda_t^2 \frac{\left( \sum_{j=1}^t (1 - \alpha_j) g_{j,i} 2^{j-t} \right)^2}{\sum_{j=1}^t (1 - \beta_j) g_{j,i}^2 2^{j-t}} \tag{38}$$

$$< 2L \sum_{t=1}^T \sum_{i=1}^d \lambda_t^2 \sum_{j=1}^t 2^{j-t} < 4dL \sum_{t=1}^T \lambda_t^2 < 4dL \lambda^2 (1 + \log T)$$

Since $\sum_{j=1}^T \frac{1}{j} \leq 1 + \log T$ and $\lambda_t = \frac{\lambda}{\sqrt{t}}$, the final inequality holds.

Substituting (38) into (37) yields,

$$\sum_{t=1}^T \langle A_t, g_t \rangle < \frac{G_2^2 d}{4L} + (f(x_1; \xi) - f(x_*)) + \lambda G_2^2 \left\| v_1^{-1/2} \right\|_2 + 4dL\lambda^2 (1 + \log T) \tag{39}$$

$$< \frac{G_2^2 d}{4L} + f(x_1; \xi) - f(x_*) + \lambda G_2^2 \left\| v_0^{-1/2} \right\|_2 + 4dL\lambda^2 (1 + \log T)$$

where $v_{0,i}^{-1/2} > v_{1,i}^{-1/2}$ and $\lambda G_2^2 > 0$.

Now we sum the inequality from equation (30) over $t = 1, ...T$ and substitute (39) into it, we obtain

$$\sum_{t=1}^{T} \left\langle \lambda_{t-1} v_{t-1}^{-1/2} \nabla f(x_t; \xi), g_t \right\rangle < \sum_{t=1}^{T} \langle A_t, g_t \rangle + \sum_{t=1}^{T} G_2^2 \left( \left\| \lambda_{t-1} v_{t-1}^{-1/2} \right\|_2 - \left\| \lambda_t v_t^{-1/2} \right\|_2 \right)$$

$$< \frac{G_2^2 d}{4L} + f(x_1; \xi) - f(x_*) + \lambda G_2^2 \left\| v_0^{-1/2} \right\|_2 + 4dL\lambda^2(1 + \log T) + \sum_{t=2}^{T} G_2^2 \left( \left\| \lambda_{t-1} v_{t-1}^{-1/2} \right\|_2 - \left\| \lambda_t v_t^{-1/2} \right\|_2 \right)$$

$$= \frac{G_2^2 d}{4L} + f(x_1; \xi) - f(x_*) + \lambda G_2^2 \left\| v_0^{-1/2} \right\|_2 + 4dL\lambda^2(1 + \log T) + G_2^2 \left( \left\| \lambda_0 v_0^{-1/2} \right\|_2 - \left\| \lambda_T v_T^{-1/2} \right\|_2 \right)$$

$$\tag{40}$$

Using $\lambda_0 = \lambda$ and $\|\lambda_T v_T^{-1/2}\|_2 \geq 0$, we have

$$\sum_{1}^{T} \left\langle \lambda_{t-1} v_{t-1}^{-1/2} \nabla f(x_t; \xi), g_t \right\rangle < \frac{G_2^2 d}{4L} + f(x_1; \xi) - f(x_*) + \lambda G_2^2 \left\| v_0^{-1/2} \right\|_2 + 4dL\lambda^2(1 + \log T) + G_2^2 \left( \left\| \lambda_0 v_0^{-1/2} \right\|_2 - \left\| \lambda_T v_T^{-1/2} \right\|_2 \right)$$

$$< \frac{G_2^2 d}{4L} + f(x_1; \xi) - f(x_*) + \lambda G_2^2 \left\| v_0^{-1/2} \right\|_2 + 4dL\lambda^2(1 + \log T) + \lambda G_2^2 \left\| v_0^{-1/2} \right\|_2$$

$$= \frac{G_2^2 d}{4L} + f(x_1; \xi) - f(x_*) + 2\lambda G_2^2 \left\| v_0^{-1/2} \right\|_2 + 4dL\lambda^2(1 + \log T)$$

$$\tag{41}$$

Since $\mathbb{E}_t$ is conditioned on the history until selecting $g_t$, $v_{t-1}$ does not depend on $g_t$, $\|v_t\|_2 \leq G_2^2$, and $\mathbb{E}_t[g_t] = \nabla f(x_t; \xi)$, we get

$$\mathbb{E}_t \left[ \left\langle \lambda_{t-1} v_{t-1}^{-1/2} \nabla f(x_t; \xi), g_t \right\rangle \right] = \left\langle \lambda_{t-1} v_{t-1}^{-1/2} \nabla f(x_t; \xi), \nabla f(x_t; \xi) \right\rangle = \sum_{i=1}^{d} \frac{\lambda_{t-1}}{v_{t-1,i}^{1/2}} (\nabla f(x_t; \xi))_i^2$$

$$< \frac{\lambda}{\sqrt{T} G_2} \|\nabla f(x_t; \xi)\|_2^2$$

$$\tag{42}$$

Taking the full expectation above yields

$$\frac{\lambda}{\sqrt{T} G_2} \mathbb{E} \left[ \|\nabla f(x_t; \xi)\|_2^2 \right] < \mathbb{E} \left[ \left\langle \alpha_{t-1} \hat{v}_{t-1}^{-1/2} \nabla f(x_t), \hat{g}_t \right\rangle \right]$$

$$< \frac{G_2^2 d}{4L} + f(x_1; \xi) - f(x_*) + 2\lambda G_2^2 \left\| v_0^{-1/2} \right\|_2 + 4dL\lambda^2(1 + \log T)$$

$$\tag{43}$$

Rearranging the above formula, we have

$$\frac{1}{T} \sum_{t=1}^{T} \mathbb{E} \left[ \|\nabla f(x_t; \xi)\|_2^2 \right] < \frac{G_2}{\lambda \sqrt{T}} \left( \frac{G_2^2 d}{4L} + f(x_1; \xi) - f(x_*) + 2\lambda G_2^2 \left\| v_0^{-1/2} \right\|_2 + 4dL\lambda^2(1 + \log T) \right).$$

$$\tag{44}$$

# 6 SIMULATION EXPERIMENTS AND DISCUSSIONS

Due to space constraints, please refer to the Appendix for details of our experiments and most experimental results. Here we only list the experimental results of the proposed algorithm on a typical graph clustering problem.

The Dual Correlation Reduction Network (DCRN) stands as a self-supervised deep graph clustering method, incorporating a dual information correlation reduction mechanism aimed at diminishing information correlation between the sample and feature levels. In accordance with the methodology outlined in Liu et al. (2022), our evaluation of optimization algorithms, such as AdamE, AdaBelief, AdaBound, AdamW, RAdam, Adam, EAdam and Padam, hinges on four widely accepted public metrics measuring clustering performance: accuracy (ACC), normalized mutual information (NMI), average rand index (ARI), and macro F1-score (F1).

The initial learning rates for AdamE, AdaBelief, AdaBound, AdamW, RAdam, Adam, and EAdam are uniformly set at 0.001, while for Padam, the initial learning rate is established as 0.01. The training of DCRN aligns with the prescribed protocol in the original literature, spanning 400 epochs until convergence. Notably, a learning rate adjustment is implemented, reducing it by a factor of 0.1 at the 50th epoch.

The experimental findings reveal significant variations in four widely acknowledged public metrics assessing clustering performance across different optimizers, as illustrated in Fig. 5(a), Fig. 5(b),

Fig. 5(c), and Fig. 5(d). Notably, the overall trend observed for AdamE in all four clustering performance metrics indicates a sustained improvement beyond epoch 250, surpassing the performance of other optimizers.

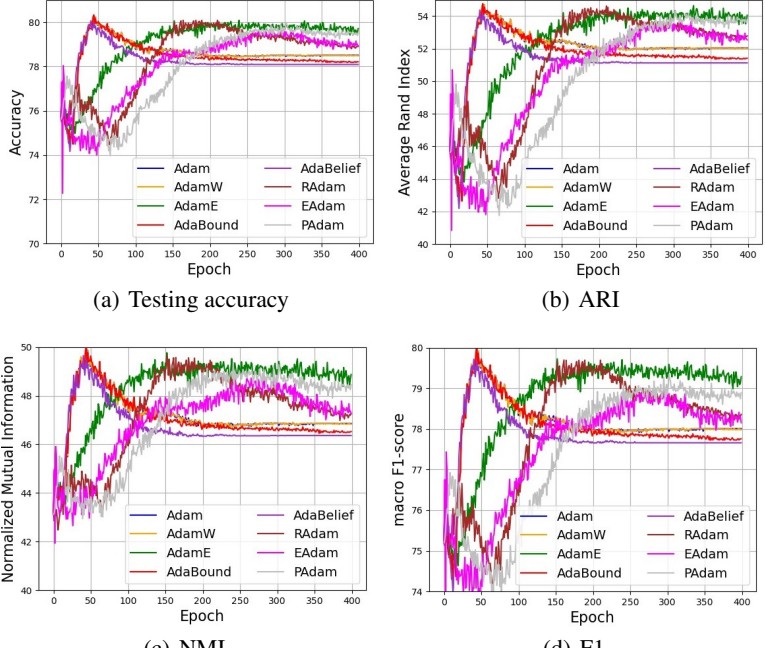

(a) Testing accuracy

(b) ARI

(c) NMI

(d) F1

Figure 5: Performance evaluation of DCRN ($\lambda = 10$) with different optimizers on DBLP.

## 7 CONCLUSIONS

In this study, we systematically analyze the impact of varying exponential decay rates on the convergence and generalization performance of the Adam optimization algorithm. Building on this analysis, we introduce an enhanced version of the algorithm, termed AdamE, and provide a comprehensive exposition of its design. The effectiveness of AdamE is demonstrated through experiments on the quadratic function fitting problem, along with a theoretical investigation of its convergence properties in convex settings.

We further evaluate the empirical performance of AdamE across a range of deep learning tasks, including language modeling, node classification, and graph clustering. Both the theoretical analyses and experimental results consistently highlight the superior performance of the proposed AdamE algorithm.

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
