# ADAPTIVE EXPONENTIAL DECAY RATES FOR ADAM-APPENDIX

## 1 SIMULATION EXPERIMENTS AND DISCUSSIONS

### 1.1 SETUP OF EXPERIMENTS

In this section, we undertake an assessment of the efficiency and effectiveness of AdamE. Our experimental approach involves the utilization of various tasks, including language modeling, node classification and graph clustering. To gauge the performance of AdamE, we compare it with several optimization algorithms, namely AdaBelief, AdaBound, AdamW, RAdam, Adam, EAdam and Padam. Diverse architectural models, such as LSTM, ProtGNN+GIN Zhang et al. (2022), Graph-MLP Hu et al. (2021), DCRN Liu et al. (2022), and UniSAGE Huang & Yang (2021), are employed, along with different learning rates, to ensure the robustness and stability of the results. All experiments are executed within the PyTorch 1.7 framework, utilizing NVIDIA Quadro RTX 8000 GPUs. The source code of LSTM, ProtGNN+GIN, Graph-MLP, DCRN, UniSAGE can be found in the Appendix 8.4.

Table 1: The data-sets and architectures used in our experiments

| Data-set | Architecture | Task |
|---|---|---|
| WikiText-2 | LSTM | Language Modeling |
| BBBP | ProtGNN+GIN | Node Classification |
| Cora | Graph-MLP | Node Classification |
| Citeseer | UniSAGE | Node Classification |
| DBLP | DCRN | Graph Clustering |

### 1.2 EXPERIMENTS ON LANGUAGE MODELING

We carry out experiment on language modeling task and consider 1-layer, 2-layer, 3-layer and 4-layer LSTM network on the WikiText-2 dataset for validating the performance of AdamE. For experiments of language modeling, the weight decay is set as $1.2 \times 10^{-6}$ for all optimizers and train LSTM network for 200 epochs with batch-size 20. At $100th$ epoch and $145th$ epoch, the learning rates is multiplied by 0.1. The initial learning rate of Adam, AdamW, AdaBound and RAdam is set as 0.001. The initial learning rate of AdaBelief and EAdam is set as 0.01, and the initial learning rate of AdamE and PAdam is set as 0.1. The results of the experiment for perplexity on the test set in Fig. 1(a), Fig. 1(b), Fig. 1(c), and Fig. 1(d), the perplexity of AdamE is lower than other methods. Moreover, the convergence of AdamE is better than other methods as in acceleration methods and good accuracy.

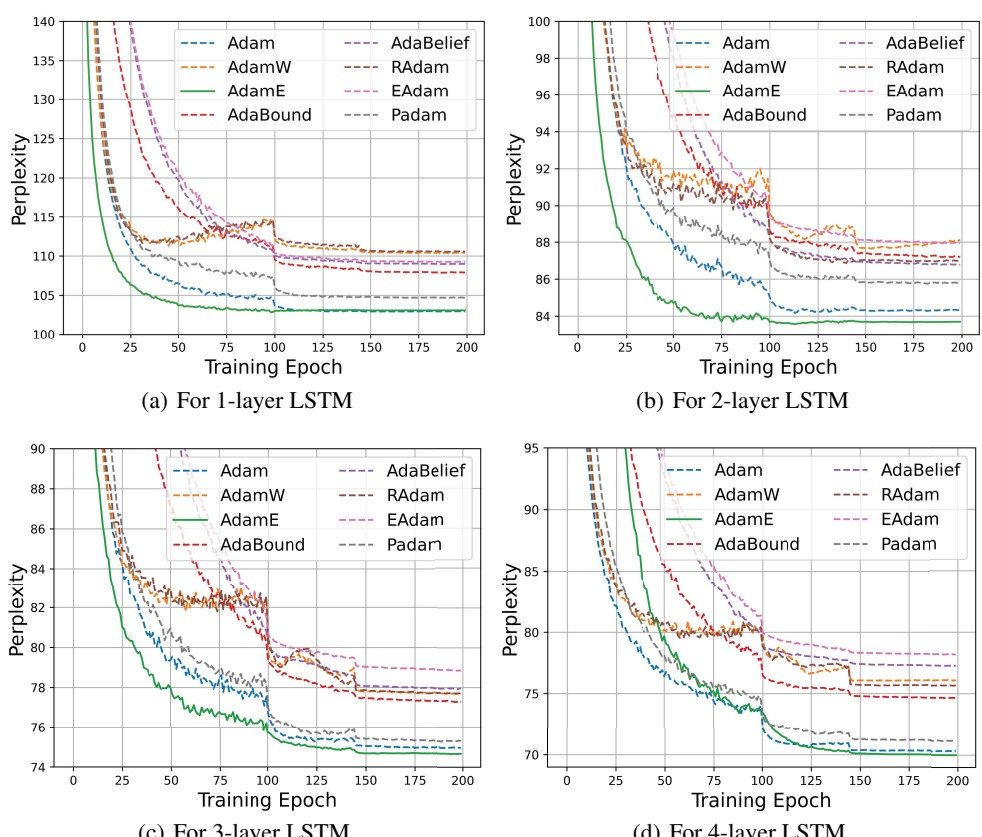

Figure 1: Test perplexity on WikiText-2.

## 1.3 EXPERIMENTS ON NODE CLASSIFICATION

Prototype GNN is formed by merging three parts: GNN encoder, prototype layer and the fully connected layer. The conditional subgraph sampling module is incorporated in ProtGNN + GIN to output subgraphs most similar to each learned propotype. The hyperparameters of ProtGNN + GIN are set as Clst=0.01 and Sep=0.05. Following Zhang et al. (2022), we adopt AdamE, AdaBelief, AdaBound, AdamW, RAdam and Adam, EAdam and Padam optimizers with fixed learning rate of 0.05, and the split for train, validation and test of BBBP dataset are set as 80%, 10%, 10%. The model of ProtGNN + GIN is trained for 100 epochs. The experiment results is shown in Fig. 2. We can see that the testing accuracy curve value fluctuations for all optimizers, but the performance of AdamE is better than other optimizers after 50 epoch.

Graph-MLP includes 256 hidden layer nodes for each linear layer which is shown in Hu et al. (2021). The activation function of Graph-MLP is Gelu, and the weighting coefficient $\alpha$ to balance the two losses is 0.6. For Graph-MLP, the initial leaning rates of AdaBelief, PAdam, AdamE and EAdam are set to 0.01, and the initial leaning rates of AdaBound, AdamW, RAdam and Adam are set to 0.001. The number of epochs is 600, and get average test accuracy of 10 epoch. The attenuation occurs every 100 iterations, and the attenuation ratio is 0.1. The results are represented in Fig. 3. Compared with other optimizers, AdamE is fast converge speed before epoch 30. The test accuracy of AdamE, AdamW, AdaBound and Adam is close after epoch 30. The overall performance of AdamE is the best.

Following Huang & Yang (2021), UniSAGE is the variant of GraphSAGE with general aggregating function (such as LSTM aggregator) and is naturally generalized as $\tilde{x}_i = W(x_i + AGGREGATE(\{h_e\}_{e \in E_i}))$. UniSAGE employs the SUM function for second-stage aggregation. We reiterate experiment of node classification over 10 data divide into 8 different random seeds. The learning rate of Adam, AdamW and AdaBound is 0.001, and the learning rate of AdamE, Ad-

aBelief, RAdam and EAdam is 0.01. 0.1 is set as the learning rate of Padam. The weight decay of all optimizers is 0.001. As shown in Fig. 4, convergence speed of AdamE is better than other optimizers before epoch 80. After epoch 100, testing accuracy on Citeseer for UniSAGE for all optimizers some volatility and the testing accuracy is approaching except for Adam.

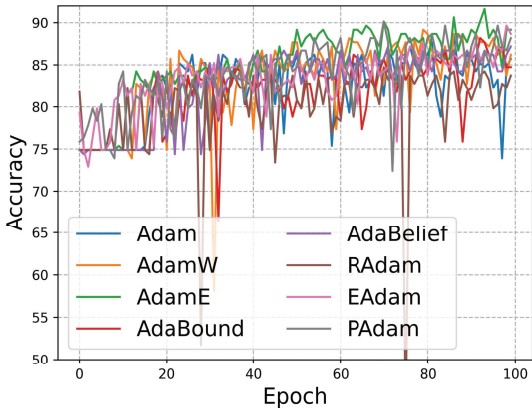

Figure 2: Testing accuracy on BBBP for ProtGNN+GIN.

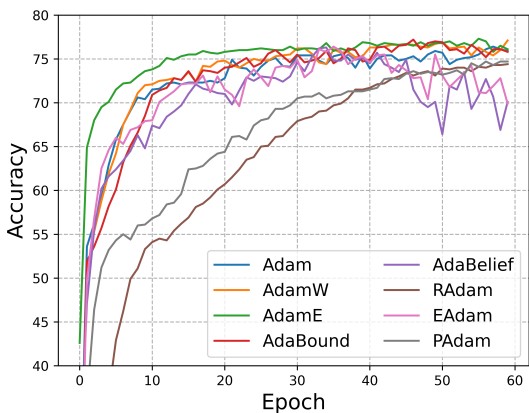

Figure 3: Testing accuracy on Cora for Graph-MLP.

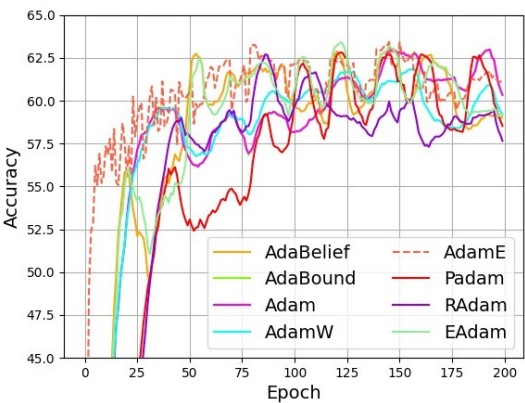

Figure 4: Testing accuracy on Citeseer for UniSAGE.