# OpenReview forum: "Adaptive Exponential Decay Rates for Adam"
_ICLR.cc/2025/Conference — Submitted to ICLR 2025_

### Official Review · Reviewer_xBZT · 2024-10-17

**Soundness:** 2
**Presentation:** 1
**Contribution:** 1
**Rating:** 3
**Confidence:** 4

**Summary:**

In this paper, the authors propose a new version of Adaptive optimization algorithm called AdamE. Instead of fixed exponential decay rates of the first and second moments, AdamE adopts an adaptive exponential decay coefficients $\alpha_t$ and $\beta_t$ for the first and second moments respectively. The authors provide both the theoretical convergence guarantee and empirical results of this new method.

**Strengths:**

This paper provides both empirical results and theoretical convergence guarantee for their new method AdamE. The theoretical results cover both convex case and non-convex case. The empirical results demonstrate that AdamE indeed performs well on clustering data. Besides, by introducing a simple example of quadratic objective, the readers can easily get the intuition of this method.

**Weaknesses:**

1. The quadratic objective example for illustration in Section 3, while easy to understand, is not suitable for comparing the performance of adaptive methods, resulting that all illustration and comparison in Section 3 is not convincing. The reasoning is quite clear: since the objective function is a scalar, different algorithms have only two choices for updating direction: positive or negative. Additionally, as a convex function, the quadratic objective has a single global minimum, denoted as $x^*$. As the iterates $x_t$ of optimizers approach $x^*$, the current gradient $g_t$ will also approach 0. Since the Adam relies heavily on historical information for its updates, its updating $\frac{m_t}{\sqrt{v_t}+\epsilon}$ cannot adapt to a small value immediately as $x_t$ approaching $x^*$. In contrast, the coefficients $\alpha_t$ and $\beta_t$ of AdamE will converge to 0 and 1, allowing it to gradually rely more on the current gradient and behave similarly to gradient descent to some extent.  Therefore, it is evident that AdamE can outperform Adam in this particular example, and I am confident that gradient descent would also perform significantly better in this simple scenario. However, I believe this does not necessarily imply that GD or AdamE is inherently superior to Adam in all cases.

2. This paper exhibits a lack of novelty in its technical contributions. Specifically, the proof for convex case is almost same with [1]. For the non-convex case, the proof sketch and main steps are also similar with [2].

3. There might exist major technical incorrectness in the proof of this paper. Firstly, in Theorem 2.2, the authors claim that the regret achieves $O(\sqrt{T})$ bound, which might be incorrect. The second term of formula (14) explicitly includes a factor of $\sqrt{T}$, while $\\|g_{1:T,i}\\|_2$ implicitly contains a factor of $\sqrt{T}$ as it is a $\ell_2$ norm of a $T$ dimensional vector.  Consequently, the second term of formula (14) is actually of order $O(T)$ instead of $O(\sqrt{T})$. In contrast, there is no explicit factor $\sqrt{T}$ in the second term of Theorem 10.5 in [1]. Secondly, for the proof of non-convex case, the authors claim a fact that $x\_{t+1}-x\_t = \lambda\_tv\_t^{-1/2}m\_t$, which is incorrect. The authors omit the existence of the stability constant $\epsilon$. In comparison, the stability constant $\epsilon$ exists in the convergence results of [2], while it disappears in this paper.

4. The writing of this paper is relatively hard to follow. Firstly, although I understand that typos are inevitable in any written work, there exist too many in this paper, rendering the mathematical derivations hard to read. For example, the $<$ in formulas (29) and (42) should be $>$. Besides, the proof lacks necessary explanations for some complex steps. In particular, during the proof of the convex case, the authors fail to use the indices from previous formulas to clarify how they derive the next formula, despite providing about 20 indices.  Furthermore, there exists significant inconsistence of notations. In the algorithm description, the authors use $d_q$ and $s_q$ to denote the first and second moments, while in the proof, it seems that they use $m_t$ and $v_t$ instead. I also suggest that authors can use some bold notations to clearly remind the readers the variables are scalars, vectors, or matrices. Finally, I suggest that the authors separate the proof of Theorem 3.1 into several lemmas to enhance clarity and understanding.

[1] Diederik P Kingma and Jimmy Ba. Adam: A method for stochastic optimization. arXiv preprint arXiv:1412.6980, 2014

[2] Dongruo Zhou, Jinghui Chen, Yuan Cao, Yiqi Tang, Ziyan Yang, and Quanquan Gu. On the convergence of adaptive gradient methods for nonconvex optimization. arXiv preprint arXiv:1808.05671, 2018.

**Questions:**

Could the author explain why $\lambda\_{t-1}v\_{t-1,i} \geq \lambda\_{t}v\_{t,i}$ holds on line 356?

---

### Official Review · Reviewer_WjW1 · 2024-11-03

**Soundness:** 1
**Presentation:** 2
**Contribution:** 1
**Rating:** 1
**Confidence:** 4

**Summary:**

This paper introduces AdamE, a variant of the Adam optimizer that adapts the exponential decay rates dynamically (based on the number of training steps) rather than relying on the default static values (e.g.,  $\beta_1 = 0.9$ ,  $\beta_2 = 0.999$ ). By adjusting decay rates based on the first and second moment estimates of gradients, AdamE aims to enhance convergence speed and overall performance in training deep neural networks. The authors provide theoretical convergence proofs in both convex and non-convex settings and validate AdamE’s performance through extensive experiments on various neural network tasks, including language modeling, node classification, and graph clustering.

**Strengths:**

The idea that using an adaptive approach to adjusting decay rates in Adam is quite novel and original, which addresses a notable challenge with existing Adam variants that rely on fixed hyperparameters, if it succeeds. However I have concerns about the correctness of the theoretical claims in this paper and also about the insufficiency of experiments. Please refer to the weakness.

**Weaknesses:**

**Correctness:** I have concern about the main theoretical results in the paper.
 - Though the authors claim they achieve a $O(\sqrt{T})$ regret bound in Theorem 2.2, Equation (14) indeed contains a linear regret term --- $\sqrt{T} \|g_{1:T,i}\|_2$.
 - In the proof of non-convex convergence case, from equation (42) to (43), the authors seem to flip the sign of the inequality by mistake.

**Lack of motivation**: Section 2 doesn't provide a convincing explanation on the motivation on why AdamE should make Adam optimizes faster. From the 1d experiments, it seems all hyper choices converge pretty fast.

**Lack of comparison to Adagrad**: When $t$ gets large, AdamE proposed in this paper essentially becomes AdaGrad, where $\alpha_q \to 0$ and $\beta_q\approx 1/q$.

**Insufficient Experiments**:  The authors only provide experiments on a few relative toy settings. I would like to see experiments on more standard benchmarks and architectures, e.g. resnet trained on Imagenet and transformers trained on common language datasets.

**Questions:**

See weakness.

---

### Official Review · Reviewer_4qKE · 2024-11-04

**Soundness:** 2
**Presentation:** 2
**Contribution:** 1
**Rating:** 3
**Confidence:** 4

**Summary:**

Instead of using (0.9,0.999)-type constant coefficients to update $m$ and $v$ in Adam, the authors propose to use a specially designed sequence for updating. The authors prove the convergence of the newly proposed algorithm in online convex setting and non-convex setting. The algorithm is tested on the DCRN network for the r graph clustering task.

**Strengths:**

The authors consider replacing the constant coefficients with a sequence of coefficients, which can help the algorithm perform better. Meanwhile, the authors give a theoretical analysis of the proposed algorithm.

**Weaknesses:**

1. In [1], they prove that when $\alpha$ and $\beta$ follow certain conditions, the Adam algorithm can converge. In [1], they have already considered the sequence coefficient with more general results.

2. There is some error in the proof of the convex setting.

(i) $||\theta_n - \theta_m\|_2 \leq D_2$ can not lead to $\|\theta_t - \theta^*\|_2 \leq D_2$. To prove this, one should first prove that $\theta_t \rightarrow \theta^*$, where the definition of $\theta^*$ is the optimal solution not the limit point of a sequence of $\theta_t$

(ii) In equation (18), all of the second terms should be $m_{t,i}^2/v_{t,i}$ instead of $m_{t,i}^2/\sqrt{v_{t,i}}$.

(iii) How to reduce $\sum_t \sqrt{t v_{t,i}}$ to $\sqrt{T,v_{T,i}}$?

[1] Zou, Fangyu, Li Shen, Zequn Jie, Weizhong Zhang, and Wei Liu. "A sufficient condition for convergences of adam and rmsprop." In Proceedings of the IEEE/CVF Conference on computer vision and pattern recognition, pp. 11127-11135. 2019.

**Questions:**

1. Do results in [1] imply the theoretical result of AdamE? If not, discuss the difference between two results.

2. Are the errors in the proof typos? If not, give the correct version of the proof.

---

### Official Review · Reviewer_H8Ku · 2024-11-08

**Soundness:** 1
**Presentation:** 1
**Contribution:** 1
**Rating:** 3
**Confidence:** 5

**Summary:**

The paper proposes a new variant of Adam that automatically tunes the $beta_1$ and $\beta_2$ of Adam.

**Strengths:**

The topic is relevant to the theme of ICLR.

**Weaknesses:**

**Weakness: most results in the script are already well-known in the literature, but not properly discussed.  I did not find much new results in this script.** I elaborate as follows.


1. many results & experiments are already discussed in the literature, but not cited. For instance

   "In this study, we explore the effects of different combinations of exponential decay rates (β1 ∈ {0.5, 0.7, 0.9} and β2 ∈ {0.9, 0.95, 0.999}) on Adam’s performance in terms of these three aspects. "

   "The experimental outcomes for Adam, .., emphasize the critical role of appropriately setting β1 and β2 for Adam based on specific tasks in training DNNs."

    The above  discussions on beta1 and beta2 have been extensively studied in [1]. Some other important works on the theory of Adam are also not cited, such as [2].

2. The proposed AdamE uses decreasing beta1 and increasing beta2. Similar method is already studied in AdamNC in [2] and [3]. I do not see any new theoretical insights in this work.  Further, the convergence analysis requires strong assumptions such as bounded gradient. Note that these types of assumptions have already been removed in the Adam analysis in [1], [4], and [5].

3. The experiments are restricted to toy settings. The practical impact is limited.



[1] Zhang, Y., Chen, C., Shi, N., Sun, R., & Luo, Z. Q. (2022). Adam can converge without any modification on update rules.

[2] Reddi, S. J., Kale, S., & Kumar, S. (2019). On the convergence of adam and beyond.

[3] Zou, F., Shen, L., Jie, Z., Zhang, W., & Liu, W. (2019). A sufficient condition for convergences of adam and rmsprop.

[4] Li, H., Rakhlin, A., & Jadbabaie, A. (2024). Convergence of adam under relaxed assumptions.

[5] Wang, B., Fu, J., Zhang, H., Zheng, N., & Chen, W. (2024). Closing the gap between the upper bound and lower bound of Adam's iteration complexity.

**Questions:**

See above

---

### Meta-Review · Area_Chair_1MSf · 2024-12-20

**Metareview:**

This paper introduces AdamE, a variant of the Adam optimizer that dynamically adapts the exponential decay rates based on the number of training steps, rather than relying on constant values such as $\beta_1 = 0.9$ and $\beta_2 = 0.999$. The method aims to improve convergence speed and overall performance by adjusting these momentum parameters.

While the paper offers a rigorous theoretical analysis of AdamE from an optimization perspective, it suffers from several notable weaknesses. The writing is difficult to follow, reducing the paper’s overall readability. Additionally, the experimental validation is weak and limited, failing to demonstrate the practical utility of the proposed method convincingly. From a methodological standpoint, the idea is relatively straightforward, and as multiple reviewers pointed out, there are concerns about the correctness of the theoretical proofs. These issues undermine the credibility of the paper’s claims and conclusions.

Given these significant limitations, I recommend rejecting this paper.

**Additional Comments On Reviewer Discussion:**

During the rebuttal period, all reviewers maintained a negative stance on the paper. The primary concerns included issues with the correctness of the theoretical proofs, the limited and weak experimental validation, and the paper’s overall readability. The authors did not provide effective responses to address these concerns, and as a result, the reviewers upheld their initial judgments to reject the paper.

---

### Decision · Program_Chairs · 2025-01-22

Reject